# CONTEXT PARAMETRIZATION WITH COMPOSITIONAL ADAPTERS

## ABSTRACT

Large language models (LLMs) often seamlessly adapt to new tasks through in-context learning (ICL) or supervised fine-tuning (SFT). However, both of these approaches face key limitations: ICL is inefficient when handling many demonstrations, and SFT incurs training overhead while sacrificing flexibility. Mapping instructions or demonstrations from context directly into adapter parameters offers an appealing alternative. While prior work explored generating adapters based on a single input context, it has overlooked the need to integrate multiple chunks of information. To address this gap, we introduce COMPAS, a meta-learning framework that translates context into adapter parameters with a compositional structure. Adapters generated this way can be merged algebraically, enabling instructions, demonstrations, or retrieved passages to be seamlessly combined without reprocessing long prompts. Critically, this approach yields three benefits: lower inference cost, robustness to long-context instability, and establishes a principled solution when input exceeds the model's context window. Furthermore, COMPAS encodes information into adapter parameters in a reversible manner, enabling recovery of input context through a decoder, facilitating safety and security. Empirical results on diverse multiple-choice and extractive question answering tasks show that COMPAS outperforms ICL and prior generator-based methods, especially when scaling to more inputs. Our work establishes composable adapter generation as a practical and efficient alternative for scaling LLM deployment.

## 1 INTRODUCTION

Large language models (LLMs) adapt to new tasks by integrating *contextual information* pertaining to these tasks, such as instructions, demonstrations, or extracted evidence. This adaptability is typically realized through in-context learning (ICL) (Brown et al., 2020), instruction following (Ouyang et al., 2022), and retrieval-augmented generation (Lewis et al., 2020; Borgeaud et al., 2022), where extra tokens in the prompt act as a transient memory that steers model behavior. Another common strategy is to adapt parameters directly through *supervised fine-tuning* (SFT), ranging from full model updates (Devlin et al., 2019; Raffel et al., 2020) to parameter-efficient variants Pfeiffer et al. (2023). While effective, both prompting and SFT face limitations. Prompt-based methods often require long contexts, which destabilize attention and degrade performance as length increases (Liu et al., 2023; Kossen et al., 2023), while also incurring inference costs that scale with context size. SFT, on the other hand, requires additional training overhead and lacks flexibility across tasks.

A complementary line of work views *adapters* (Houlsby et al., 2019; Hu et al., 2022) as a mechanism for parametrizing context. In our work, we use *context* as an umbrella term for instructions, demonstrations, or supporting passages. By translating context into parameters, adapters offer a persistent representation that replaces prompt tokens, thereby improving the stability of processing long contexts and amortizing adaptation across tasks (Karimi Mahabadi et al., 2021; He et al., 2022; Liu et al., 2023). Once context is encoded into adapters, inference requires only the query tokens, reducing the cost of processing long prompts and making the approach especially attractive in latency- or memory-constrained settings. This context-to-adapter transformation encodes an otherwise transient token sequence into a manipulable object, allowing for caching, reuse, and algebraic combination.

Building on this perspective, recent work has leveraged meta-learning to efficiently generate adapter parameters directly from support examples without fine-tuning anew for each context (Zhang et al.,

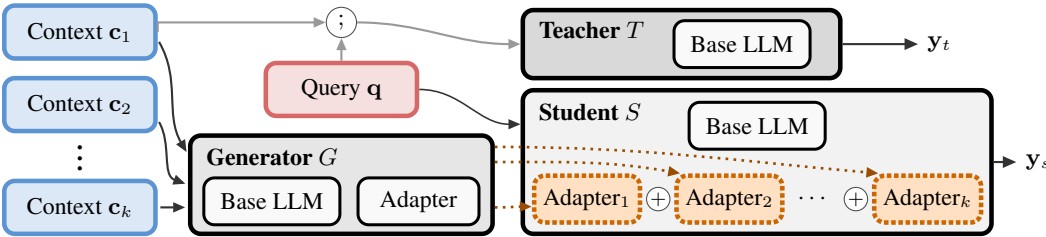

Figure 1: Overview of the COMPAS framework. Each context $c_i$ is mapped into an adapter by the generator $G$, while the query $q$ is processed by the student $S$. The teacher $T$ processes the concatenated input $[c; q]$, and $G$ is trained so that composed adapters in $S$ align $y_s$ with $y_t$.

2023; Wang et al., 2023; Chen et al., 2025). These works largely focused on producing adapters from a single *context*. In practice, however, contexts rarely occur in isolation. Tasks are often guided by chains of instructions, multiple demonstrations, or sets of retrieved passages that must be integrated. Without a principled notion of composition, such adapters cannot faithfully capture how LLMs combine multiple sources of context. Composability also addresses the limitations of prompt-based adaptation. Instead of concatenating long contexts, adapters can be merged algebraically in parameter space. The key challenge is therefore not parameter generation alone, but algebraically consistent composition that preserves semantics of concatenated texts while retaining efficient inference.

To bridge this critical gap, we introduce COMPAS (**Comp**ositional **A**dapter**s**), a framework that maps context into adapter parameters with compositional structure at its core. See Figure 1 for a high-level overview. In COMPAS, a teacher LM process concatenated contexts, while the student LM learns to approximate this behavior using only query tokens, complemented with the sum of adapters generated for individual contexts. This is facilitated by a generator network, which maps individual contexts into compositional adapters. COMPAS is driven by auxiliary compositionality and reconstruction losses, which provide diagnostics for whether parameter-space operations preserve semantics and faithfulness (Jacovi & Goldberg, 2020; Turpin et al., 2023; Min et al., 2023; Lampinen et al., 2022). We evaluate COMPAS on multiple-choice and extractive QA tasks, where either demonstrations (MMLU, ARC-Challenge) or retrieved passages (HotpotQA) serve as context. We observe consistent improvements over ICL, fine-tuning and existing context-to-parameter methods. These gains are most pronounced when scaling to larger numbers of demonstrations, highlighting the capacity of COMPAS for efficient integration of external evidence by composition in parameter-space.

Our contributions are threefold: (1) We introduce COMPAS, a teacher-student framework for encoding context into adapters which facilitates composition in parameter space. (2) We establish theoretical conditions under which parameter-space addition provably approximates the behavior of the teacher model. (3) We empirically show that COMPAS outperforms alternatives, especially when scaling to large context sizes, demonstrating the benefits of compositional integration of information.

## 2 ADAPTER COMPOSITIONALITY

Our goal is to ensure that adapters generated from individual contexts can be *composed* in parameter space to replicate the effect of context concatenation in input space. More concretely, in a teacher–student setup, the outputs of the teacher LM produced when given multiple contexts jointly should be reproduced by the student LM using only the query tokens along with the sum of the corresponding adapter parameters. We now formalize this requirement, introducing conditions under which adapter addition in the parameter space corresponds to context concatenation in the input space.

### 2.1 SETUP

Let $V$ be the vocabulary and $|V|$ its size. A language model with frozen parameters $\theta \in \mathbb{R}^n$ is a function $f_\theta : \mathcal{X} \to \mathbb{R}^{|V|}$, where $\mathcal{X} = \Sigma^*$ is the set of all token sequences over the alphabet $\Sigma$. For $x \in \mathcal{X}$, the output $f_\theta(x) \in \mathbb{R}^{|V|}$ denotes the logit vector over the vocabulary. A *support context* $c \in \mathcal{C}$ is any textual information (e.g., a demonstration, instruction, or retrieved passage) provided

alongside a query $\mathbf{q} \in \mathcal{X}$, where $\mathcal{C} \subseteq \mathcal{X}$ denotes the set of admissible contexts. We use $[\mathbf{c}; \mathbf{q}] \in \mathcal{X}$ to denote the concatenation of a context $\mathbf{c}$ with the query $\mathbf{q}$. Let $\Phi \subseteq \mathbb{R}^m$ denote the set of adapter parameters, and let $\phi \in \Phi$ denote parameters for a single adapter, i.e., a structured modification of $\boldsymbol{\theta}$. We write $\boldsymbol{\theta} \oplus \phi$ for the parameters obtained by composing $\boldsymbol{\theta}$ with $\phi$.

The **teacher** model corresponds to the base LM applied to the concatenated input: $\mathbf{f}_T(\mathbf{c}, \mathbf{q}) :=$ $\mathbf{f}_{\boldsymbol{\theta}}([\mathbf{c}; \mathbf{q}])$. The **student** model processes only the query, under parameters modified by the adapter: $\mathbf{f}_S^{\phi}(\mathbf{q}) := \mathbf{f}_{\boldsymbol{\theta} \oplus \phi}(\mathbf{q})$. For a sequence contexts $\mathbf{c}_1, \ldots, \mathbf{c}_k$, the teacher takes the concatenation $[\mathbf{c}_1; \ldots; \mathbf{c}_k; \mathbf{q}]$, while the student uses the query together with the summed adapter $\sum_{i=1}^{k} \phi_i$. For example, with two contexts, $\mathbf{c}_1$ and $\mathbf{c}_2$, the student reduces to $\mathbf{f}_S^{\phi_1 + \phi_2}(\mathbf{q}) = \mathbf{f}_{\boldsymbol{\theta} \oplus (\phi_1 + \phi_2)}(\mathbf{q})$. Finally, a **generator** $G : \mathcal{C} \to \Phi$ maps each context $\mathbf{c} \in \mathcal{C}$ into an adapter $\phi = G(\mathbf{c})$. In particular, for a pair of contexts $\mathbf{c_1}$ and $\mathbf{c_2}$ we define $\phi_1 := G(\mathbf{c}_1)$, $\phi_2 := G(\mathbf{c}_2)$, and $\phi_{12} := G([\mathbf{c}_1; \mathbf{c}_2])$.

## 2.2 COMPOSITIONALITY BOUND

The set of finite contexts $\mathcal{C}$, equipped with concatenation and the empty string identity $(\lambda)$, forms a free monoid $(\mathcal{C}, [\cdot; \cdot], \lambda)$. On the other hand, the adapter space $(\Phi, +, \mathbf{0})$ is a commutative additive monoid. A fundamental requirement for adapter compositionality is that the generator $G : \mathcal{C} \to \Phi$ be a monoid homomorphism, $G([\mathbf{c}_1; \mathbf{c}_2]) = G(\mathbf{c}_1) + G(\mathbf{c}_2)$ and $G(\lambda) = \mathbf{0}$, so that addition in the adapter space exactly mirrors concatenation in the input space. Demanding $G$ to be a homomorphism aligns these two structures. Concatenation on $\mathcal{C}$ is associative but not commutative, whereas addition on $\Phi$ is both. Thus, requiring $G$ to be a homomorphism implicitly collapses the order of contexts. This means that $G$ is a non-injective homomorphism: permuted concatenations map to the same adapter sum. We regard this loss of order as desirable in many cases, as the contribution of a demonstration or retrieved passage should often be independent of its relative position. By treating contexts as a set rather than a sequence, we aim for adapters that focus on semantics, mitigating the order sensitivity and instability observed in long-context transformers.[1]

Enforcing a homomorphism between the free monoid of contexts and the additive monoid of adapter parameters via a teacher–student setup cannot be achieved exactly in practice, given the models' finite representational capacity, the finiteness of the training sample, and the approximate nature of optimization. We therefore approximate it, which introduces discrepancies. To capture these discrepancies and regularity conditions, we introduce three quantities.

**Student–teacher error.** For a context $\mathbf{c}$ with the corresponding adapter $\phi = G(\mathbf{c})$, the student's discrepancy from the teacher is

$$\epsilon(\mathbf{c}) := \|\mathbf{f}_S^{\phi}(\mathbf{q}) - \mathbf{f}_T([\mathbf{c}; \mathbf{q}])\|_2, \tag{1}$$

**Generator additivity error.** The generator's deviation from additivity is

$$\eta := \|G([\mathbf{c}_1; \mathbf{c}_2]) - (G(\mathbf{c}_1) + G(\mathbf{c}_2))\|_2. \tag{2}$$

**Parameter sensitivity.** The student's logits vary smoothly with respect to adapter parameters, with a Lipschitz constant

$$L := \sup_{\phi_1, \phi_2, \mathbf{q}} \frac{\|\mathbf{f}_S^{\phi_1}(\mathbf{q}) - \mathbf{f}_S^{\phi_2}(\mathbf{q})\|_2}{\|\phi_1 - \phi_2\|_2}. \tag{3}$$

The first two quantities capture the core requirements of compositionality, while the third provides a regularity condition needed for the analysis. Assuming $L < \infty$, i.e., Lipschitz continuity of the student with respect to adapter parameters, we obtain the following result.

**Theorem 1** (Compositionality Bound)**.** *For any contexts* $\mathbf{c}_1, \mathbf{c}_2$*, and query* $\mathbf{q}$*, with* $\phi_i = G(\mathbf{c}_i)$*,*

$$\|\mathbf{f}_S^{\phi_1 + \phi_2}(\mathbf{q}) - \mathbf{f}_T([\mathbf{c}_1; \mathbf{c}_2; \mathbf{q}])\|_2 \leq L\eta + \epsilon([\mathbf{c}_1; \mathbf{c}_2]).$$

---

[1]Certain tasks may still require positional information (e.g., reasoning with ordered evidence). We show in Appendix C.3 that our framework can incorporate explicit positional encodings when needed.

See Appendix A for the proof. The bound decomposes the student–teacher error into two interpretable sources: generator additivity error ($\eta$) and misfit on the concatenated context ($\epsilon([\mathbf{c}_1; \mathbf{c}_2])$). The Lipschitz constant $L$ propagates generator errors into the student. Thus, perfect compositionality is unattainable in practice, but approximate compositionality is achievable whenever these quantities remain small.

## 3 METHOD

Previously, we decomposed the discrepancy between *context concatenation* and *adapter addition* into two main error sources: generator additivity error ($\eta$) and student–teacher error on concatenated contexts ($\epsilon$), with parameter sensitivity ($L$) controlling how generator errors propagate to outputs. While $L$ is a structural property of the student model, we can *design* the generator and its training objectives to directly minimize $\eta$ and $\epsilon$. To this end, we introduce COMPAS, a teacher–student meta-learning framework that maps each support context into a LoRA adapter (Hu et al., 2022), which is composed additively with the base parameters. The adapter generator ends with a linear bottleneck, ensuring that its outputs in parameter space add directly. This algebraic addition is trained to approximate the *outputs* of the model under context concatenation in input space. To achieve this, we design loss functions that incentivize the generator to approximate a homomorphism from context concatenation to parameter-space addition.

### 3.1 CONTEXT-TO-ADAPTER GENERATOR

As introduced in Section 2, the generator $G$ maps a context $\mathbf{c}$ to adapter parameters $\phi = G(\mathbf{c})$. It is implemented by augmenting the base LM with additional trainable components: (i) its own LoRA adapter inserted into the LM, and (ii) a linear bottleneck followed by two projection trunks. The same frozen base LM is shared across the teacher, the student, and the generator, where only the generator's adapter and projection components are updated during training.

Given a support context $\mathbf{c}$, the base LM (with the generator's adapter) processes the tokens, and we obtain $\mathbf{h}(\mathbf{c}) \in \mathbb{R}^d$ as the mean of the final-layer hidden states across all tokens. This pooled representation is projected into a compact latent space,

$$\mathbf{z}(\mathbf{c}) = P\,\mathbf{h}(\mathbf{c}), \qquad P \in \mathbb{R}^{r \times d}, \ \ r \ll d,$$

reducing dimensionality and ensuring the subsequent mapping into LoRA parameters remains tractable. Without this bottleneck, a direct mapping would require prohibitively many parameters.

A LoRA module applied to a target linear map with input dimension $d^{\text{in}}$ and output dimension $d^{\text{out}}$ consists of two low-rank matrices $\mathbf{A} \in \mathbb{R}^{r \times d^{\text{in}}}$ and $\mathbf{B} \in \mathbb{R}^{d^{\text{out}} \times r}$, with rank $r \ll d^{\text{in}}, d^{\text{out}}$. Each transformer layer may contain multiple such modules (e.g., query, key, value, or output projections). Let $\mathcal{M}$ denote the set of all modules instrumented with LoRA, indexed across layers and projection types. From the latent representation $\mathbf{z}(\mathbf{c})$, the generator produces parameters for all modules via two bias-free linear projections: $U_A \mathbf{z}(\mathbf{c}) \in \mathbb{R}^{m_A}$ and $U_B \mathbf{z}(\mathbf{c}) \in \mathbb{R}^{m_B}$, with

$$m_A = \sum_{m \in \mathcal{M}} r_m d_m^{\text{in}}, \qquad m_B = \sum_{m \in \mathcal{M}} d_m^{\text{out}} r_m.$$

The outputs of $U_A$ and $U_B$ are partitioned and reshaped into the individual LoRA matrices $\mathbf{A}_m(\mathbf{c})$ and $\mathbf{B}_m(\mathbf{c})$ for each target module $m \in \mathcal{M}$. See Figure 1 for a high-level overview of COMPAS.

### 3.2 LOSS FUNCTION COMPONENTS

Our loss function design follows Theorem 1: we reduce the student–teacher error in both single and concatenated contexts, and penalize generator non-additivity. We also include a reconstruction term that encourages adapters to faithfully encode their contexts (up to permutation), preventing collapse into trivial solutions. This auxiliary objective also regularizes training and provides a stronger learning signal, aiding optimization.

**Student–teacher alignment.** We draw unlabeled queries $\mathbf{q}$ and contexts of length $k \in \{1, 2\}$, letting the teacher provide soft pseudo-labels (logits). The student is trained to match these through

*weak supervision*:

$$\mathcal{L}_{\mathrm{ST}} = \mathbb{E}_{((\mathbf{c}_1,\ldots,\mathbf{c}_k),\,\mathbf{q})}\Big[\mathrm{KL}\Big(\mathrm{softmax}\big(\mathbf{f}_T([\mathbf{c}_1;\ldots;\mathbf{c}_k;\mathbf{q}])\big) \,\big\|\, \mathrm{softmax}\big(\mathbf{f}_S^{\sum_{i=1}^k \phi_i}(\mathbf{q})\big)\Big)\Big].$$

where KL denotes Kullback–Leibler divergence. For $k = 1$ this enforces single-context fidelity, while for $k = 2$ it enforces compositionality.

**Additivity regularization.** We reduce the generator additivity error $\eta$ by penalizing discrepancies between parameters generated for concatenated contexts and the sum of parameters from the individual contexts. Concretely, we draw pairs of contexts $(\mathbf{c}_1, \mathbf{c}_2)$ and compare generated parameters:

$$\mathcal{L}_{\mathrm{ADD}} = \mathbb{E}_{(\mathbf{c}_1,\mathbf{c}_2)}\left[\sum_{m\in\mathcal{M}} \frac{\big\|\mathbf{W}_m([\mathbf{c}_1;\mathbf{c}_2]) - \big(\mathbf{W}_m(\mathbf{c}_1) + \mathbf{W}_m(\mathbf{c}_2)\big)\big\|_F^2}{\big\|\mathbf{W}_m([\mathbf{c}_1;\mathbf{c}_2])\big\|_F^2 + \delta}\right],$$

where $\mathbf{W}_m(\mathbf{c}) \in \{\mathbf{A}_m(\mathbf{c}), \mathbf{B}_m(\mathbf{c})\}$ denotes either of the two LoRA matrices generated for module $m \in \mathcal{M}$, $\|\cdot\|_F$ is the Frobenius norm, and $\delta > 0$ is a small constant for stability.

**Reconstruction.** To encourage faithfulness and improve training stability, we require adapters to recoverably encode the information contained in their contexts. We introduce a special query token `[RECON]`, which prompts the student to reconstruct the support context from its adapter in an autoregressive manner. This discourages collapse to trivial solutions and provides an auxiliary learning signal through the cross-entropy (CE) loss:

$$\mathcal{L}_{\mathrm{RECON}} = \mathbb{E}_{\mathbf{c}}\Big[\mathrm{CE}\big(\mathbf{c}, \mathbf{f}_S^{\phi}(\texttt{[RECON]})\big)\Big].$$

**Overall objective.** The final loss function is a weighted sum:

$$\mathcal{L}_{\mathrm{COMPAS}} = \lambda_{\mathrm{ST}}\mathcal{L}_{\mathrm{ST}} + \lambda_{\mathrm{ADD}}\mathcal{L}_{\mathrm{ADD}} + \lambda_{\mathrm{RECON}}\mathcal{L}_{\mathrm{RECON}}. \tag{4}$$

Hyperparameter choices and training details are provided in Appendix B.

## 4 EXPERIMENTS

We first outline the experimental design §4.1, with full setup and hyperparameter details in Appendix B. We then evaluate COMPAS in two regimes: (i) replacing in-context demonstrations with adapter parameters (§4.2), and (ii) encoding retrieved passages as parametric memory (§4.3). Finally, we assess reconstruction, measuring faithfulness of information encoded in adapter parameters (§4.4). Additional experiments on context order sensitivity and efficiency are in Appendix C.

### 4.1 EXPERIMENTAL SETUP

**Models.** We experiment with decoder-only LMs: **LLaMA-3.1 8B** (LLaMA 8B) and **Llama-3.2 3B** (Dubey et al., 2024), and **Qwen-2.5 7B** (Yang et al., 2025). For brevity, we refer to these models as LLaMA 8B, LLaMA 3B, and Qwen 7B.

**Tasks.** We consider two representative settings: (i) ICL on **MMLU** (Hendrycks et al., 2021) and **ARC-Challenge** (Clark et al., 2018), where few-shot exemplars are provided as demonstrations; (ii) Extractive question answering (QA) on **HotpotQA** (Yang et al., 2018), where gold passages supply contextual evidence. Prompts and preprocessing are standardized across all methods (Appendix D).

**Methods.** We compare against three representative approaches. **ICL** is standard $k$-shot prompting with demonstrations concatenated in prompt text. **Generative Adapter (GenAda)** (Chen et al., 2025) employs a nonlinear hypernetwork to predict adapter weights from context, trained with reconstruction and continuation losses. In contrast, our method enforces a linear bottleneck, generating adapters via a lightweight adapter module followed by a linear transformation. WILDA (Jukić & Šnajder, 2025) fine-tunes a separate adapter for each context, achieving strong accuracy but incurring heavy computational overhead, since a new adapter must be trained for every context.

Table 1: Demonstration parameterization results. Rows with a single number indicate concatenation of all demonstrations without composition, while rows of the form $a \times b$ denote composition of $a$ adapters, each encoding $b$ demonstrations. Results are reported as the mean over 10 runs with the standard deviation as a subscript. The best score within each block is highlighted in **bold**. COMPAS results are statistically compared to the corresponding ICL setting; scores marked with † indicate significance under a Wilcoxon signed-rank test ($p < 0.05$) with Holm–Bonferroni correction.

| Setup / Method | | Llama-3.1 8B | | Llama-3.2 3B | | Qwen-2.5 7B | |
|---|---|---|---|---|---|---|---|
| | | MMLU | ARC | MMLU | ARC | MMLU | ARC |
| 4 | ICL | $64.2_{1.8}$ | $74.2_{1.6}$ | $57.2_{1.8}$ | $62.9_{1.6}$ | $70.2_{1.8}$ | $76.5_{1.6}$ |
| 4 | GenAda | $57.6_{1.8}$ | $69.3_{1.7}$ | $50.4_{1.4}$ | $59.4_{1.7}$ | $59.8_{2.1}$ | $70.5_{1.7}$ |
| 4 | WILDA | $\mathbf{68.8}_{0.9}$ | $\mathbf{78.2}_{0.7}$ | $\mathbf{60.8}_{1.1}$ | $\mathbf{66.4}_{1.2}$ | $\mathbf{74.0}_{0.4}$ | $\mathbf{79.6}_{0.9}$ |
| 4 | COMPAS | $66.7_{0.7}{}^{\dagger}$ | $76.5_{0.8}{}^{\dagger}$ | $59.0_{0.9}$ | $65.1_{0.8}{}^{\dagger}$ | $72.4_{0.5}{}^{\dagger}$ | $77.2_{0.6}$ |
| 8 | ICL | $65.5_{1.7}$ | $75.1_{1.5}$ | $58.0_{1.7}$ | $64.1_{1.5}$ | $71.5_{1.7}$ | $78.3_{1.5}$ |
| 8 | GenAda | $59.1_{1.9}$ | $69.9_{1.6}$ | $51.3_{1.9}$ | $61.2_{1.6}$ | $60.7_{1.9}$ | $71.4_{1.6}$ |
| 8 | WILDA | $\mathbf{69.8}_{0.7}$ | $78.8_{0.8}$ | $61.2_{1.1}$ | $\mathbf{67.4}_{0.5}$ | $\mathbf{74.8}_{0.7}$ | $80.0_{0.8}$ |
| 8 | COMPAS | $67.5_{0.5}$ | $77.2_{0.9}{}^{\dagger}$ | $60.2_{1.3}{}^{\dagger}$ | $66.2_{0.5}{}^{\dagger}$ | $73.2_{1.2}$ | $78.6_{0.7}$ |
| 2×4 | GenAda | $57.8_{1.9}$ | $69.5_{1.6}$ | $50.9_{0.7}$ | $59.4_{1.2}$ | $60.5_{1.8}$ | $69.2_{1.6}$ |
| 2×4 | WILDA | $69.5_{0.3}$ | $78.6_{0.9}$ | $60.8_{1.1}$ | $67.1_{1.0}$ | $74.1_{0.4}$ | $79.8_{0.8}$ |
| 2×4 | COMPAS | $69.1_{0.7}{}^{\dagger}$ | $\mathbf{79.3}_{0.9}{}^{\dagger}$ | $\mathbf{61.3}_{1.2}{}^{\dagger}$ | $66.8_{0.6}{}^{\dagger}$ | $74.6_{0.6}{}^{\dagger}$ | $\mathbf{80.2}_{0.9}$ |
| 12 | ICL | $66.0_{1.7}$ | $75.5_{1.5}$ | $58.8_{1.7}$ | $64.6_{1.5}$ | $71.8_{1.7}$ | $77.2_{1.5}$ |
| 12 | GenAda | $59.9_{1.9}$ | $70.2_{1.6}$ | $51.6_{1.9}$ | $62.5_{1.6}$ | $63.1_{1.9}$ | $71.6_{1.2}$ |
| 12 | WILDA | $70.5_{0.8}$ | $79.4_{0.6}$ | $61.9_{1.1}$ | $\mathbf{68.1}_{1.0}$ | $75.2_{0.7}$ | $79.8_{0.4}$ |
| 12 | COMPAS | $67.2_{1.2}$ | $78.9_{0.9}{}^{\dagger}$ | $59.5_{1.2}$ | $67.2_{0.7}{}^{\dagger}$ | $72.9_{1.0}$ | $79.3_{1.4}$ |
| 3×4 | GenAda | $57.3_{1.9}$ | $70.3_{1.6}$ | $49.8_{1.9}$ | $59.6_{1.6}$ | $62.0_{1.9}$ | $69.7_{1.6}$ |
| 3×4 | WILDA | $70.7_{1.0}$ | $79.5_{0.9}$ | $62.0_{1.1}$ | $67.5_{1.0}$ | $75.1_{0.7}$ | $79.9_{0.4}$ |
| 3×4 | COMPAS | $\mathbf{71.2}_{1.1}{}^{\dagger}$ | $\mathbf{80.1}_{0.5}{}^{\dagger}$ | $\mathbf{62.4}_{0.8}{}^{\dagger}$ | $67.4_{0.9}$ | $\mathbf{75.7}_{0.3}{}^{\dagger}$ | $\mathbf{80.3}_{0.8}{}^{\dagger}$ |
| 16 | ICL | $66.5_{1.7}$ | $76.0_{1.5}$ | $59.2_{1.7}$ | $65.2_{1.5}$ | $72.3_{1.7}$ | $77.6_{1.5}$ |
| 16 | GenAda | $60.5_{1.9}$ | $72.5_{1.6}$ | $52.7_{1.2}$ | $60.8_{1.3}$ | $64.2_{1.4}$ | $71.9_{0.9}$ |
| 16 | WILDA | $71.5_{0.7}$ | $80.4_{0.3}$ | $62.6_{0.5}$ | $68.2_{0.3}$ | $76.1_{0.4}$ | $80.7_{0.8}$ |
| 16 | COMPAS | $68.2_{0.6}$ | $79.1_{0.7}{}^{\dagger}$ | $62.3_{1.2}$ | $68.3_{0.4}{}^{\dagger}$ | $73.7_{0.6}$ | $79.4_{0.5}$ |
| 4×4 | GenAda | $57.6_{1.9}$ | $70.6_{1.6}$ | $50.3_{1.9}$ | $59.9_{1.6}$ | $61.3_{1.9}$ | $70.4_{1.6}$ |
| 4×4 | WILDA | $71.6_{0.6}$ | $80.5_{0.9}$ | $62.7_{1.1}$ | $68.3_{1.1}$ | $75.2_{0.5}$ | $80.8_{0.8}$ |
| 4×4 | COMPAS | $\mathbf{72.2}_{0.3}{}^{\dagger}$ | $\mathbf{81.3}_{0.3}{}^{\dagger}$ | $\mathbf{63.4}_{0.7}{}^{\dagger}$ | $\mathbf{69.3}_{0.5}{}^{\dagger}$ | $\mathbf{77.1}_{0.7}{}^{\dagger}$ | $\mathbf{81.5}_{0.4}{}^{\dagger}$ |

**Evaluation.** We assess models on end-task performance, measuring accuracy on MMLU and ARC and exact match (EM) and token-level F1 on HotpotQA. Faithfulness is evaluated through the KL divergence between the teacher concatenation and the student sum, as well as token-level F1 (*bag-of-tokens*). Stability is measured as the standard deviation across ten runs with different sampled contexts. Training follows the weakly supervised protocol from §3.2. Full implementation details, including optimization settings and hyperparameters, are deferred to Appendix B.

## 4.2 ENCODING DEMONSTRATIONS AS PARAMETERS

We first evaluate COMPAS effectiveness in replacing in-context demonstrations with adapter parameters. In the $k$-shot setting ($k \in \{4, 8, 12, 16\}$), demonstrations are partitioned into fixed-size blocks, each block is encoded as an adapter, and the adapters are composed by summing their parameters. Table 1 shows the results on MMLU and ARC-Challenge across three base models.

COMPAS *consistently outperforms standard ICL*, which in our setup corresponds to using the teacher model directly, across all models and shot counts, while also exhibiting lower variance. We attribute this robustness to weak-to-strong (W2S) generalization (Dherin et al., 2022; Lang et al., 2024): the student begins with weak pseudo-labels from the teacher but progressively corrects them during training, extending reliability from easy, locally consistent regions to harder examples.

Gains are most pronounced at higher shot counts ($k \in 12, 16$) under 3×4 and 4×4 composition settings, where COMPAS consistently outperforms WILDA (11 out of 12 cases), despite the latter's

Table 2: Extractive QA results on single, pair, and triplet gold contexts, averaged over 10 runs and reported as EM/F1. *Teacher (concat)* evaluates the base LLM directly on concatenated gold contexts (e.g., $[c_1; c_2; c_3]$). COMPAS (*concat*) evaluates the student using an adapter generated from the concatenated context, $G([c_1; c_2; c_3])$, while COMPAS (*composed*) uses the sum of adapters from individual contexts, $\sum_i G(c_i)$.

| Setup | Method | LLaMA-3.1 8B | LLaMA-3.2 3B | Qwen-2.5 7B |
|---|---|---|---|---|
| **Single (c)** | Teacher | 70.9/82.0 | 65.9/78.1 | 69.2/80.7 |
| | COMPAS | 72.2/82.9 | 67.0/78.6 | 69.4/82.2 |
| **Pair ($[c_1; c_2]$)** | Teacher (concat) | 69.0/80.8 | 64.1/76.8 | 67.5/79.8 |
| | COMPAS (concat) | 69.4/81.1 | 64.5/77.0 | 67.9/79.5 |
| | COMPAS (composed) | 71.3/82.3 | 66.2/78.4 | 69.6/80.9 |
| **Triplet ($[c_1; c_2; c_3]$)** | Teacher (concat) | 66.3/78.5 | 61.2/74.1 | 64.0/76.0 |
| | COMPAS (concat) | 67.5/79.4 | 62.0/75.3 | 65.1/77.0 |
| | COMPAS (composed) | 71.1/82.0 | 64.8/76.9 | 68.1/80.4 |

Table 3: COMPAS context reconstruction as token-level F1. Results are averages over 10 runs. MMLU and ARC *units* are blocks of 4 demonstrations; a HotpotQA *unit* is a supporting paragraph.

| | Llama-3.1 8B | | | Llama-3.2 3B | | | Qwen-2.5 7B | | |
|---|---|---|---|---|---|---|---|---|---|
| **Setup** | **MMLU** | **ARC** | **Hotpot** | **MMLU** | **ARC** | **Hotpot** | **MMLU** | **ARC** | **Hotpot** |
| Single (1 unit) | 91.2 | 89.5 | 87.8 | 89.0 | 87.2 | 85.4 | 90.1 | 88.1 | 86.2 |
| Pair (2 units) | 90.3 | 88.9 | 86.0 | 88.1 | 86.3 | 83.5 | 89.4 | 87.5 | 84.3 |
| Triplet (3 units) | 88.4 | 86.9 | 83.7 | 86.5 | 84.8 | 81.9 | 87.6 | 85.9 | 82.5 |

use of context-specific adapter fine-tuning. We hypothesize that this advantage arises from an implicit chunking–composition mechanism, where the model partitions demonstrations into manageable subsets. Each adapter encodes a partial context view, and their addition reconstructs the effect of all demonstrations. As a result, COMPAS scales gracefully with the number of demonstrations: it leverages composition to maintain strong performance in long contexts, ultimately matching or exceeding specialized adapter fine-tuning while retaining efficiency and composability.

### 4.3 ENCODING CONTEXT AS PARAMETRIC MEMORY

We now evaluate COMPAS on extractive QA by treating gold evidence passages as *parametric memories*. On HotpotQA, each query $q$ is paired with one or more gold passages $c$. When multiple passages are available (pairs or triplets), the task becomes more challenging: the model must leverage a larger combined context and answer more queries. In these settings, we either generate a single adapter from their concatenation or compose the adapters generated from each passage individually. We evaluate per query, including all gold passages in either form. Results are shown in Table 2.

COMPAS consistently outperforms the teacher (base LLM) across all settings. Absolute performance is lower on pairs and triplets – since the model must retain context for multiple queries while being evaluated on each one separately – but the relative gains over the teacher are larger in these harder settings. This indicates that COMPAS is particularly effective at composing and retaining information across multiple contexts.

### 4.4 CONTEXT RECONSTRUCTION

Finally, we test whether composed adapters preserve the informational content of their supports by prompting the model to reconstruct missing contexts. Using a special `[RECON]` token, we decode from adapters and compute token-level F1 on MMLU, ARC, and HotpotQA. Table 3 shows that COMPAS maintains consistently high reconstruction fidelity, with only small drops as the context expands to pair and triplet units.

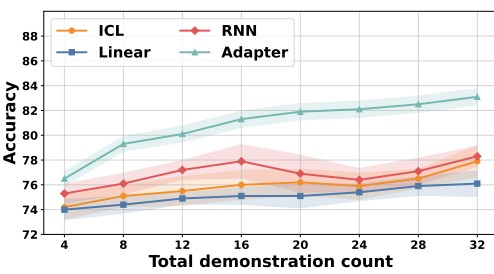 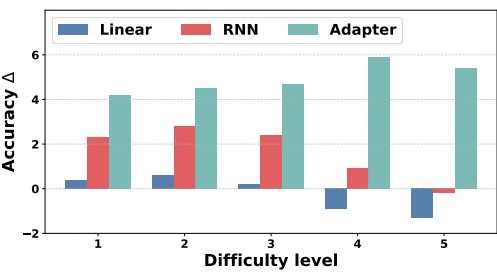

Figure 2: Effect of generator capacity on accuracy with LLaMA 8B on ARC-Challenge. Shaded areas show deviation over 10 runs.

Figure 3: Accuracy deltas of different generators over ICL across five MMLU difficulty buckets (average over 10 runs for LLaMA 8B).

## 5 ANALYSIS

Building on the effectiveness of COMPAS, we analyze the role of generator capacity (§5.1) and the contribution of each loss component (§5.2). We discuss limitations of our approach in Appendix E.

### 5.1 CAPACITY AND WEAK-TO-STRONG GENERALIZATION

We explore three generator capacities $G$: **Adapter** (COMPAS) – corresponds to our default setup (§3); **RNN** – a lightweight recurrent network which aggregates token-level encodings of $\mathbf{c}$ before the same linear head produces $G(\mathbf{c})$; **Linear** – an ablation that removes the generator adapter and recurrent aggregator, leaving a single bottleneck linear projection of a pooled context representation (see configuration details in Appendix B).

In Figure 2, we compare the performance of generators with varying capacity as the number of demonstrations increases. A simple linear generator initially tracks the performance of the ICL teacher, but falls behind as more demonstrations are added, indicating limited ability to capture multi-shot composition. The *RNN* generator improves on ICL and linear baselines for moderate demonstration counts, but eventually stagnates. In contrast, the *Adapter* generator consistently outperforms alternatives, and its advantage widens with larger demonstration counts. These results highlight the importance of generator capacity for both surpassing the ICL teacher and maintaining compositionality as context scales.

In Figure 3, we analyze the effects of W2S generalization by grouping MMLU tasks into five difficulty levels (1–5). To obtain these groups, we evaluate the base LLaMA 8B with 16 shots on each MMLU subtask and sort them by accuracy, partitioning into five bins. We then evaluate generators in a $4\times4$ setup (16 demonstrations in total). Across levels 1–3, all generator variants improve over ICL, with gains ordered by capacity (*Adapter*, *RNN*, and *Linear*). At higher difficulty, differences sharpen: the linear generator collapses at level 4–5, *RNN* stagnates and drops at level 5, while *Adapter* continues to improve, achieving the largest gains on the hardest tasks. These results show that sufficient generator capacity is crucial for surpassing the ICL teacher and maintaining additive composition as complexity grows. COMPAS scales with task difficulty, enhancing W2S generalization on the hardest tasks.

### 5.2 IMPORTANCE OF LOSS COMPONENTS

To better understand the contribution of each training signal, we perform an ablation study on the loss components of COMPAS. Table 4 reports results on MMLU, ARC-Challenge, and HotpotQA using the LLaMA 8B backbone. We compare the full objective (4) to variants with removed components, isolating their effect on overall performance.

We always include $\mathcal{L}_{ST}$, since weak supervision from the teacher is essential for transferring contextual information into the student. Dropping $\mathcal{L}_{ADD}$ or $\mathcal{L}_{RECON}$ degrades performance and shows that they play complementary roles. $\mathcal{L}_{ADD}$ reduces additivity error ($\eta$) and stabilizes multi-context composition. $\mathcal{L}_{RECON}$ provides smaller but consistent gains while also enabling context reconstruction. Removing both terms ($\mathcal{L}_{ADD}$ and $\mathcal{L}_{RECON}$) leads to the most pronounced drop, confirming that

Table 4: Ablation of loss function components. Results are on MMLU and ARC ($4\times4$ demos) and HotpotQA (EM/F1, pairs). We report averages over 10 seeds with standard deviations as subscripts.

| Objective Variant | MMLU | ARC-Challenge | HotpotQA (EM/F1) |
|---|---|---|---|
| **Full** | 69.1 | 79.3 | 71.3/82.3 |
| $-\mathcal{L}_{\text{ADD}}$ | 64.8 | 75.7 | 68.2/79.3 |
| $-\mathcal{L}_{\text{RECON}}$ | 66.4 | 77.1 | 70.2/80.8 |
| $\mathcal{L}_{\text{ST}}$ only | 61.2 | 71.8 | 65.9/76.4 |

they jointly support the effectiveness of the student. Overall, the full objective yields the strongest results, consistent with the theory in §2.

## 6 RELATED WORK

**ICL and parameterized adaptation.** Analyses show ICL often exploits surface-level cues rather than learning task semantics (Min et al., 2022), and that transformers can internally implement simple learning algorithms during inference (Xie et al., 2021; Akyürek et al., 2022). In parallel, *parameter-efficient fine-tuning* (PEFT) replaces long prompts with compact trainable modules such as soft/prefix prompts (Li & Liang, 2021b; Lester et al., 2021) or low-rank adapters (Hu et al., 2021). Recently, Hong et al. (2024) proposed mixtures of in-context learners to efficiently combine subsets of demonstrations. Like PEFT, our work keeps inference on the query only, but instead of concatenating demonstrations, we translate them into compositional adapters.

**Composing and merging parameters.** Another line of research explores how to *compose* learned modules or entire models. In the adapter setting, AdapterFusion learns to fuse multiple task adapters non-destructively (Pfeiffer et al., 2021), while MAD-X composes language and task adapters for cross-lingual transfer (Pfeiffer et al., 2020). For full model weights, *model soups* average fine-tuned checkpoints to improve robustness (Wortsman et al., 2022), and *task vectors* use weight differences to add or subtract task behaviors (Ilharco et al., 2022). These approaches combine pre-trained modules or models after the fact. In contrast, we generate adapters *on-the-fly* from context and train them explicitly for additivity, so that parameter-space composition mirrors text concatenation.

**Generating parameters from context.** *HyperNetworks* generate parameters of one network using another (Ha et al., 2017). Ansell et al. (2021) introduced MAD-G, where a hypernetwork generates language adapters conditioned on language embeddings. Subsequent work extended this idea to few-shot adaptation (Zhang et al., 2023; Chen et al., 2025). Our method can be viewed as a hypernetwork explicitly designed for *compositionality*: a generator maps support encodings into LoRA parameters, while compositional distillation and a linearity regularizer enforce that adapter addition in parameter space approximates context concatenation in input space. This aligns with broader efforts to impose linear additive structure on representations (Sulc, 2025), but uniquely emphasizes parameter-space composition as a scalable route to context integration.

## 7 CONCLUSION

We tackled the challenges of efficiency and instability that arise when adapting LLMs to tasks requiring multiple demonstrations or retrieved passages. To address these issues, we introduced COMPAS, a meta-learning teacher–student framework that translates contexts into *compositional adapters*. COMPAS encodes context into adapter parameters that can be algebraically combined, allowing processing of complex queries without input concatenation. In this way, we enable efficient handling of large context sets by generating adapters independently and composing them in parameter space, reducing inference cost, mitigating long-context degradation, and circumventing context window limitations. A reconstruction objective promotes safety and security, ensuring that the input context is decodable from the adapter parameters. COMPAS consistently outperforms ICL and prior generator-based approaches on multi-choice and extractive question answering. Taken together, these results establish composable context parametrization as a scalable approach for adapting LLMs.

## REPRODUCIBILITY STATEMENT

Along with the description of our method in Section 3 and the experimental setup in Appendix B, we also provide comprehensive details to ensure reproducibility. All datasets used (MMLU, ARC-Challenge, HotpotQA) are publicly available and described in Section 4.1. Model configurations, optimization settings, and training schedules are fully specified in Appendix B, along with the specified computing infrastructure. Theoretical results, including the proof of Theorem 1 and its corollary, are presented in Appendix A. Prompt templates for all benchmarks are provided in Appendix D. In addition, we include code in the supplementary material and will publish the repository publicly after the review process. Collectively, these resources allow independent reproduction and verification of our experiments and analyses.

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

## A   COMPOSITIONALITY BOUND

### A.1   PROOF OF THEOREM 1

Let $\phi_1 = G(\mathbf{c}_1)$, $\phi_2 = G(\mathbf{c}_2)$, and $\phi_{12} = G([\mathbf{c}_1; \mathbf{c}_2])$. We aim to bound

$$\left\| \mathbf{f}_S^{\phi_1 + \phi_2}(\mathbf{q}) - \mathbf{f}_T([\mathbf{c}_1; \mathbf{c}_2; \mathbf{q}]) \right\|_2.$$

The Euclidean space $(\mathbb{R}^{|V|}, \|\cdot\|_2)$ satisfies the triangle inequality: for any $a, b, c \in \mathbb{R}^{|V|}$,

$$\|a - c\|_2 \;\leq\; \|a - b\|_2 \;+\; \|b - c\|_2, \tag{5}$$

where we also use absolute homogeneity (symmetry) of the norm, i.e., $\|b - c\|_2 = \|c - b\|_2$ since $\|-v\|_2 = \|v\|_2$. Apply (5) with

$$a = \mathbf{f}_S^{\phi_1 + \phi_2}(\mathbf{q}), \quad b = \mathbf{f}_S^{\phi_{12}}(\mathbf{q}), \quad c = \mathbf{f}_T([\mathbf{c}_1; \mathbf{c}_2; \mathbf{q}]),$$

to obtain

$$\|\mathbf{f}_S^{\phi_1 + \phi_2}(\mathbf{q}) - \mathbf{f}_T([\mathbf{c}_1; \mathbf{c}_2; \mathbf{q}])\|_2$$
$$\leq \underbrace{\|\mathbf{f}_S^{\phi_1 + \phi_2}(\mathbf{q}) - \mathbf{f}_S^{\phi_{12}}(\mathbf{q})\|_2}_{\text{Term (I)}} \;+\; \underbrace{\|\mathbf{f}_S^{\phi_{12}}(\mathbf{q}) - \mathbf{f}_T([\mathbf{c}_1; \mathbf{c}_2; \mathbf{q}])\|_2}_{\text{Term (II)}}. \tag{6}$$

By parameter sensitivity (3) and generator additivity error (2),

$$\text{Term (I)} \;\leq\; L\,\|\phi_1 + \phi_2 - \phi_{12}\|_2 \;=\; L\,\eta. \tag{7}$$

Apply the student–teacher error (1) to the concatenated context $[\mathbf{c}_1; \mathbf{c}_2]$:

$$\text{Term (II)} \;\leq\; \epsilon([\mathbf{c}_1; \mathbf{c}_2]). \tag{8}$$

Combining (6), (7), and (8) yields

$$\|\mathbf{f}_S^{\phi_1 + \phi_2}(\mathbf{q}) - \mathbf{f}_T([\mathbf{c}_1; \mathbf{c}_2; \mathbf{q}])\|_2 \;\leq\; L\eta + \epsilon([\mathbf{c}_1; \mathbf{c}_2]),$$

which proves Theorem 1. $\qquad\square$

**Corollary 1** (Extension to $k$ contexts). *By induction, Theorem 1 extends to any sequence of contexts* $\mathbf{c}_1, \ldots, \mathbf{c}_k$ *with corresponding adapters* $\phi_i = G(\mathbf{c}_i)$:

$$\|\mathbf{f}_S^{\sum_{i=1}^{k} \phi_i}(\mathbf{q}) - \mathbf{f}_T([\mathbf{c}_1; \ldots; \mathbf{c}_k; \mathbf{q}])\|_2 \;\leq\; (k-1)L\eta \;+\; \epsilon([\mathbf{c}_1; \ldots; \mathbf{c}_k]).$$

### A.2   GEOMETRIC INTERPRETATION

Adapters $\phi = G(\mathbf{c})$ can be viewed as vectors in the parameter space $\Phi$. In the ideal case, $G$ embeds contexts into a flat subspace where concatenation in input space corresponds exactly to vector addition in parameter space. The deviation terms then acquire a geometric meaning: $\eta$ measures the generator's departure from additivity, and $\epsilon([\mathbf{c}_1; \mathbf{c}_2])$ captures the student's misalignment with the teacher on concatenated contexts. The Lipschitz constant $L$ governs how deviations in adapter space propagate into the model's output space. Compositionality can therefore be understood as flattening the contextual geometry into a nearly linear embedding, with small $\eta$ and $\epsilon$ ensuring that adapter addition faithfully mirrors context concatenation.

## B   EXPERIMENTAL SETUP AND HYPERPARAMETERS

### B.1   DATA

For MMLU (Hendrycks et al., 2021), which comprises 59 subject-specific subsets, we report accuracy averaged across all subsets. We train a *single generator* jointly across all subsets. This ensures that the generator is shared across domains rather than tuned separately per subject. For weak supervision during generator optimization, we sample queries and contexts from the validation split

Table 5: Parameter counts for base models and our framework. Student parameters correspond to LoRA adapters attached to the base LM, while generator parameters denote the generator network itself. Generator adapter counts indicate the size of parameters produced per context. Total trainable parameters are the sum of student and generator parameters.

| Model | Base LLM | Student adapter | Generator adapter | Generator (total) |
|-------|----------|-----------------|-------------------|-------------------|
| Llama 3.1 (8B) | 8.03B | 21M | 42M | 210M |
| Llama 3.2 (3B) | 3.19B | 4.2M | 8.4M | 42M |
| Qwen 2.5 (7.6B) | 7.61B | 20M | 40M | 180M |

without using ground-truth labels, while demonstrations are drawn (with labels) from the training split. Evaluation is performed on the held-out test split.

For ARC-Challenge (Clark et al., 2018), we follow the same setup: unlabeled validation examples provide queries and contexts for weak supervision, labeled demonstrations are drawn from the training set, and performance is reported on the test set.

For HotpotQA (Yang et al., 2018), we restrict context inputs to the relevant gold supporting passages associated with each question. We sample 10k examples stratified across difficulty levels, and partition them into 5k for training, 3k for validation, and 2k for testing.

## B.2 MODELS

We summarize the main characteristics of the base LMs used in our experiments in Table 5, together with the sizes of the student and generator adapters. We use the `bfloat16` half-precision format for all model parameters.

## B.3 METHODS

For GenAda (Chen et al., 2025) and WILDA (Jukić & Šnajder, 2025), we use the default hyperparameters and follow the training procedures described in the respective papers. To ensure comparability, we adopt an identical configuration for adapter sizes and target modules across all methods; these design choices are detailed in the following section.

## B.4 HYPERPARAMETERS

**Optimization.** For each dataset and model combination, we train the generator parameters with AdamW (weight decay 0.01) for 10 epochs. The learning rate is set to $10^{-4}$ with 5% linear warmup followed by cosine decay.

**Loss weights.** We set $\lambda_{\text{ST}} = 1.0$, $\lambda_{\text{ADD}} = 0.5$, and $\lambda_{\text{RECON}} = 0.1$. These values balance (i) fidelity to teacher logits through student–teacher alignment, (ii) enforcement of compositionality via additivity regularization, and (iii) auxiliary reconstruction for stability and faithfulness, while avoiding over-regularization.

**Adapter configuration.** We insert LoRA adapters into all attention projections ($q$, $k$, $v$, and $o$) as well as into the MLP down- and up-projection layers. For the generator, we use a uniform rank of 32 across all modules, while for the student we use rank 16. Following standard practice, each LoRA module applies a scaling factor $\alpha$, such that the effective weight update is $\frac{\alpha}{r}AB$. We set $\alpha = 32$ for generator adapters and $\alpha = 16$ for student adapters, ensuring balanced contribution of low-rank updates relative to their respective ranks.

**Reconstruction under composition.** In the compositional setting, where multiple adapters are generated for different contexts and summed, we compute the reconstruction F1 score in a permutation-invariant fashion. This adjustment is necessary because the generator is explicitly incentivized to be commutative, i.e., $G([c_1; c_2]) \approx G(c_1) + G(c_2) = G(c_2) + G(c_1)$. Concretely, for multiple contexts, we evaluate reconstruction across all possible permutations and report the

Table 6: XSUM summarization with LLaMA-3.1 8B. We report ROUGE-1 (R1), ROUGE-2 (R2), and ROUGE-L (RL) F1. R1 and R2 measure unigram and bigram overlap with the reference summary, while RL measures the longest common subsequence. Higher values indicate closer alignment with the reference and better summary quality.

| Method | R1 ↑ | R2 ↑ | RL ↑ |
|--------|------|------|------|
| ICL | 36.1 | 15.6 | 28.4 |
| PBFT | 38.3 | 18.1 | 31.6 |
| GenAda | 33.4 | 13.1 | 25.6 |
| COMPAS | **41.8** | **22.8** | **33.5** |

score corresponding to the most successful ordering. This ensures that the metric reflects content preservation rather than sensitivity to input order.

**Generator variants.** In addition to the default *Adapter* generator used in COMPAS, we consider two alternatives for ablation. The *RNN* variant employs a lightweight two-layer gated recurrent unit (GRU; Cho et al., 2014) with hidden size 256 to aggregate token-level context representations before projecting them into a compact latent space, followed by an up-projection to the full set of student adapter parameters. The *Linear* variant removes both the GRU and the generator adapter; instead, the pooled context representation is passed through a single linear bottleneck and then expanded directly into the student adapter parameter space.

### B.5 COMPUTING INFRASTRUCTURE

We conducted our experiments on a mix of local and cluster resources. Local training was performed on a workstation with an AMD Ryzen Threadripper 3970X 32-Core CPU, 256GB RAM, and $2\times$ NVIDIA GeForce RTX 3090 GPUs (24GB each). Larger-scale runs were executed on a compute cluster equipped with $2\times$ NVIDIA A100 GPUs (40GB each).

## C ADDITIONAL EXPERIMENTS

### C.1 LONG-FORM SUMMARIZATION

Beyond question answering, we also evaluate whether the representations learned by COMPAS generalize to long-form generation tasks. Its reconstruction objective already requires producing full demonstrations and multi-paragraph passages (Table 3), exercising long-range generation in training.

To test generalization capabilities beyond QA, we additionally evaluate COMPAS on XSUM, a standard abstractive summarization benchmark. Results are reported in Table 6. COMPAS achieves the strongest performance across all ROUGE metrics, substantially outperforming ICL and other baselines. This indicates that the learned context-to-parameter mapping transfers to tasks that require extended, structured generation, supporting the use of COMPAS as a general parametrization mechanism.

### C.2 SUPERVISED AND PROMPT-BASED BASELINES

We further evaluate supervised and parametric baselines: (i) pattern-based fine-tuning (PBFT), optimized with a cloze-style language modeling objective (Schick & Schütze, 2021); (ii) a linear SFT head trained on top of the frozen backbone; (iii) prefix-tuning with the same demonstrations (Li & Liang, 2021a); and (iv) an ICL variant that retrieves the top-$k$ most similar demonstrations for each query using SBERT embeddings (Reimers & Gurevych, 2019).

Results for MMLU and ARC are reported in Table 7. Across both benchmarks, COMPAS consistently outperforms all supervised and prompt-based baselines, supporting that its improvements arise from the context-to-parameter mechanism rather than differences in data, supervision, or training procedure.

Table 7: Supervised and prompt-based baselines with LLaMA-3.1 8B. We report mean accuracy on MMLU and ARC over 10 runs with 16 demonstrations in total. COMPAS is evaluated with $4 \times 4$ composition. Standard deviations are shown in subscript, and the best results are in bold.

| Method | MMLU | ARC |
|---|---|---|
| ICL (16-shot) | $64.2_{1.8}$ | $74.2_{1.6}$ |
| ICL + prefix-tuning | $66.1_{1.4}$ | $74.0_{1.3}$ |
| ICL + similarity selection | $68.1_{0.6}$ | $75.5_{0.2}$ |
| PBFT (labels only) | $66.8_{1.1}$ | $75.6_{1.0}$ |
| Linear SFT head | $62.1_{1.5}$ | $71.3_{1.4}$ |
| COMPAS ($4\times4$) | $\mathbf{72.2_{0.3}}$ | $\mathbf{81.3_{0.3}}$ |

Table 8: Position-aware reconstruction evaluation. We report positional ROUGE-L F1 for context pair permutations from HotpotQA for the commutative baseline without positional encoding (No-PE) and a simple position-aware variant (PE-Tag), and slot-based positional embeddings (Slot-PE). Scores are mean with standard deviation as subscripts over 10 runs.

| | LLaMA-3.1 8B | LLaMA-3.2 3B | Qwen-2.5 7B |
|---|---|---|---|
| No-PE (commutative linear) | $62.5_{2.0}$ | $50.1_{2.2}$ | $60.4_{2.1}$ |
| Slot-PE | $87.6_{0.9}$ | $71.1_{2.6}$ | $85.0_{1.4}$ |

### C.3 POSITIONAL ENCODING AND ORDER SENSITIVITY

By design, our generator enforces *commutativity*, i.e., $G([c_1; c_2]) \approx G(c_1) + G(c_2) = G(c_2) + G(c_1)$, which is desirable for tasks where order invariance is beneficial and directly reduces $\eta$. The trade-off is that strict commutativity removes the ability to encode order.

As a *proof of concept*, we conduct an experiment on HOTPOTQA using pairs of supporting contexts. We augment the generator with *slot embeddings*, i.e., for slot $i$ we add a learnable vector $s_i$ to the corresponding adapter delta:

$$\tilde{\phi}_i = G(c_i) + s_i, \qquad \Phi = \sum_i \tilde{\phi}_i. \tag{9}$$

This introduces positional variance while preserving linear composition, thereby allowing the model to distinguish $[c_1; c_2]$ from $[c_2; c_1]$.

Results in Table 8 suggest that order sensitivity can be layered on top of the commutative backbone when tasks require it, though we leave a systematic study to future work.

### C.4 COMPOSITIONAL SCALING

A central design choice in COMPAS is how context is partitioned into blocks prior to adapter generation. Block size controls both how much information is encoded into a single adapter and how many such adapters must be composed at inference time, and therefore directly influences both expressiveness and compositional stability. We next examine how this tradeoff affects accuracy and additivity as the total number of demonstrations increases.

Table 9 shows how block structure affects both task accuracy and compositional fidelity. For a fixed number of demonstrations, COMPAS reaches its best performance with intermediate block sizes (approximately 4–8 demonstrations per block), which yield both higher MMLU accuracy and lower additivity error $\eta$.

When blocks become too large (e.g., $2 \times 16$ or $1 \times 16$), the generator must encode many demonstrations jointly, which weakens the linear structure of the latent parameterization and increases additivity error. Conversely, when blocks become too small (e.g., $16 \times 1$), each block provides a limited contextual signal, leading to slightly reduced accuracy and moderately higher $\eta$.

As the total number of demonstrations increases, accuracy steadily improves, demonstrating that COMPAS benefits from additional evidence. In practice, the maximum feasible block size is limited

Table 9: Effect of block granularity and scaling on MMLU accuracy and additivity error $\eta$ with LLaMA 8B. We vary the block structure for a fixed total number of demonstrations and report accuracy and generator additivity error.

| Total demos | Method | Accuracy | $\eta \downarrow$ |
|---|---|---|---|
| 16 | ICL (16-shot) | 64.2 | – |
| 16 | COMPAS (4×4) | 72.2 | 0.08 |
| 16 | COMPAS (2×8) | 70.9 | 0.11 |
| 16 | COMPAS (1×16) | 69.4 | 0.15 |
| 16 | COMPAS (16×1) | 71.3 | 0.09 |
| 32 | ICL (32-shot) | 65.5 | – |
| 32 | COMPAS (8×4) | 73.1 | 0.07 |
| 32 | COMPAS (4×8) | 74.0 | 0.09 |
| 32 | COMPAS (2×16) | 72.4 | 0.19 |
| 40 | COMPAS (5×8) | 74.3 | 0.10 |
| 48 | COMPAS (6×8) | 74.9 | 0.09 |
| 56 | COMPAS (7×8) | 75.4 | 0.12 |
| 64 | COMPAS (8×8) | 76.2 | 0.16 |

by GPU memory, since larger blocks require the generator to encode more tokens jointly. Within these limits, using moderately sized blocks provides a favorable tradeoff between representation stability and information density.

Finally, long-context ICL baselines can only be evaluated reliably up to 32 demonstrations due to memory constraints. In contrast, COMPAS decouples inference cost from context length, enabling evaluation with more demonstrations on the same hardware through compositional adapter merging.

## C.5 ROBUSTNESS TO NOISY CONTEXT

In practical retrieval-based systems, the contexts provided to a model are often imperfect: demonstrations may contain incorrect labels, multiple sources may disagree about the same example, or retrieved passages may be unrelated to the query. We therefore evaluate how COMPAS behaves under controlled perturbations that explicitly simulate these failure modes and compare it directly to standard ICL under identical corruption patterns.

We introduce three types of degradation at inference time. *Noisy demonstrations* are generated by randomly flipping the labels of a fixed fraction of the in-context examples (12.5% and 25% of the 16 demonstrations), while keeping inputs unchanged. This simulates annotation errors or unreliable retrieval from weakly supervised corpora. *Contradictory demonstrations* are constructed by duplicating a subset of examples and assigning them conflicting labels: for 4 of the 16 demonstrations, the same input appears twice across contexts, once with the correct label and once with an incorrect label. This explicitly creates inconsistent supervision inside the context and mimics disagreement between retrieved sources or conflicting knowledge bases. *Off-topic contamination* is introduced by appending unrelated paragraphs sampled from HotpotQA to the end of the context, increasing the number of irrelevant tokens without changing the gold demonstrations. This simulates retrieval failures in which non-informative or spurious documents are mixed into the prompt. All results are reported as averages over 10 independent runs with different random corruption patterns.

Table 10 reports accuracy on MMLU as the demonstration noise increases. Both methods degrade as noise grows, but COMPAS consistently retains a clear performance margin. Notably, the gap widens under stronger label corruption and remains substantial under contradictory supervision. This suggests that compositional context encoding is more robust to internally inconsistent evidence than raw concatenation: while ICL treats all tokens uniformly, COMPAS compresses each demonstration into a latent representation, reducing the impact of spurious or conflicting signals.

Table 11 reports results on HotpotQA as irrelevant passages are injected into the context. ICL performance degrades steadily as more off-topic content is introduced, indicating sensitivity to prompt dilution and distraction effects. In contrast, COMPAS remains comparatively stable and preserves

Table 10: Robustness under noisy and contradictory demonstrations on MMLU with LLaMA 8B (mean over 10 runs).

| Method | Clean | 12.5% noise | 25% noise | Contradictory |
|--------|-------|-------------|-----------|---------------|
| ICL | 66.5 | 62.1 | 58.7 | 57.9 |
| COMPAS | **72.2** | **71.3** | **69.4** | **68.4** |

Table 11: HotpotQA under off-topic context contamination with LLaMA 8B. Accuracy with irrelevant paragraphs appended to the context (mean over 10 runs).

| Method | Clean | +1 off-topic | +2 off-topic |
|--------|-------|--------------|--------------|
| ICL | 82.0 | 79.4 | 77.6 |
| COMPAS | 82.9 | 82.3 | 81.7 |

most of its clean-context accuracy even as irrelevant text grows. This behavior is consistent with the hypothesis that encoding context into parameters rather than raw tokens makes inference less sensitive to unrelated input.

Taken together, these experiments demonstrate that composition-based context parametrization improves robustness to realistic retrieval noise. Even when demonstrations contain errors, contradictions, or irrelevant information, COMPAS degrades more gracefully than ICL, supporting additive composition as a stabilizing mechanism under imperfect and noisy context conditions.

## C.6 EFFICIENCY

We compare the computational and memory efficiency of COMPAS to standard ICL prompting using LLaMA 8B on ARC-Challenge. Table 12 reports FLOPs speedup (relative to ICL with the same number of demonstrations $k$) and peak inference memory. Results are averaged over 10 runs on a sample of 1k test queries. As expected, the memory cost of ICL grows linearly with the number of demonstrations, since the model must re-encode all tokens at inference. In contrast, COMPAS amortizes context encoding into a one-shot parameter generation step, so inference depends only on the query length. This yields substantial speedups that increase with $k$: $2.2\times$ at 4-shot, $3.7\times$ at 12-shot, and $4.1\times$ at 16-shot. Memory usage shows a similar trend, with ICL increasing steadily as more demonstrations are added, while COMPAS remains nearly constant across different $k$. This demonstrates that COMPAS not only improves accuracy, but also yields more efficient and scalable inference by decoupling compute and memory from context length.

Additionally, Table 13 compares absolute inference and training compute across methods. The methods differ primarily in their training regimes. WILDA is inexpensive for a single demonstration configuration, but does not amortize: each new context requires a separate fine-tuning run. GenAda incurs a large one-time training cost to learn a generator. COMPAS lies between these extremes, training a generator once at moderate cost and reusing it across contexts. In cumulative terms, WILDA's total cost grows linearly with the number of demonstration configurations and overtakes COMPAS after approximately 17 contexts, beyond which repeated fine-tuning becomes more expensive than the one-time generator training of COMPAS. At the same time, COMPAS is approximately $13\times$ cheaper to train than GenAda, while achieving comparable inference-time efficiency. Thus, COMPAS provides a favorable tradeoff between repeated fine-tuning and large upfront training cost, combining amortization with moderate training overhead.

Table 12: Efficiency comparison of ICL vs. COMPAS with LLaMA 8B. We report speedups in FLOPs for COMPAS inference relative to ICL with $k$ demonstrations, and peak inference memory (GB). Results are averaged over 10 runs on ARC-Challenge. COMPAS replaces long-context prompts with adapters, reducing both compute and memory usage.

| | Speedup (FLOPs rel. to ICL@$k$) $\uparrow$ | | | | Peak memory (GB) $\downarrow$ | | | |
|---|---|---|---|---|---|---|---|---|
| **Method** | **4** | **8** | **12** | **16** | **4** | **8** | **12** | **16** |
| ICL (prompting) | 1.0× | 1.0× | 1.0× | 1.0× | 22.1 | 27.3 | 33.5 | 37.9 |
| COMPAS | 2.2× | 3.1× | 3.7× | 4.1× | 17.2 | 17.9 | 18.1 | 19.4 |

Table 13: Absolute compute comparison with LLaMA 8B. We report absolute inference-time compute per query and training compute in TFLOPs on the ARC-Challenge dataset using 16 demonstrations in total with $4 \times 4$ composition for COMPAS. WILDA trains a new adapter for each demonstration configuration, whereas GenAda and COMPAS train a generator once and amortize across different contexts.

| Method | Inference TFLOPs / query | Training TFLOPs | Regime |
|---|---|---|---|
| ICL (prompting) | 4.0 | – | No training |
| WILDA | 0.7 | $1.4 \times 10^5$ | Per demo set |
| GenAda | 0.9 | $3.1 \times 10^7$ | One-time |
| COMPAS | 0.8 | $2.4 \times 10^6$ | One-time |

## D PROMPT TEMPLATES

### D.1 TEMPLATES FOR MULTI-CHOICE QUESTION ANSWERING

---

**Generic prompt template MMLU and ARC-Challenge**

**Demonstrations:**
```
Question: {Previous Question 1}
Answer choices:
  (A: {Choice A1}),
  (B: {Choice B1}),
  (C: {Choice C1}),
  (D: {Choice D1})
Answer: (Correct Answer 1)

Question: {Previous Question 2}
Answer choices:
(A: {Choice A2}),
(B: {Choice B2}),
(C: {Choice C2}),
(D: {Choice D2})
Answer: (Correct Answer 2)
...
```
**Query:**
```
Question: {Current Question}
Answer choices:
(A: {Choice A}),
(B: {Choice B}),
(C: {Choice C}),
(D: {Choice D})
Answer: (
```

---

## D.2 MMLU EXAMPLES

---

**Example for MMLU `abstract_algebra`**

**Demonstrations:**

```
Question: Find the maximum possible order for an element of S_n for
    n = 10.
Answer choices:
(A: 6),
(B: 12),
(C: 30),
(D: 105)
Answer: (C: 30)

Question: Compute the product in the given ring. (2,3)(3,5) in Z_5
    x Z_9
Answer choices:
(A: (1,1)),
(B: (3,1)),
(C: (1,6)),
(D: (3,6))
Answer: (D: (3,6))
```

**Query:**

```
Question: If (G, .) is a group such that (ab)^-1 = a^-1b^-1
for all a, b in G, then G is a/an
Answer choices:
(A: commutative semigroup),
(B: abelian group),
(C: non-abelian group),
(D: None of these)
Answer: (
```

---

## D.3 ARC-CHALLENGE EXAMPLES

---

**Example for ARC-Challenge**

**Demonstrations:**

```
Question: Based on their locations in the periodic table,
which element has chemical properties most similar
to those of calcium, Ca?
Answer choices:
(A: beryllium, Be),
(B: potassium, K),
(C: titanium, Ti),
(D: yttrium, Y)
Answer: (A: beryllium, Be)

Question: Which term best describes the life cycle of an insect
that reaches the adult stage without being a pupa?
Answer choices:
(A: incomplete metamorphosis),
(B: complete metamorphosis),
(C: alternation of generations),
(D: spontaneous mutation)
Answer: (A: incomplete metamorphosis)
```

**Query:**

```
Question: Which property of a mineral can be determined
```

---

```
just by looking at it?
Answer choices:
(A: luster),
(B: mass),
(C: weight),
(D: hardness)
Answer: (
```

## D.4 HOTPOTQA EXAMPLE

---

**Example for HotpotQA**

**Question:**
Who invented the type of script used in autographs?

**Supporting Context:**

```
(Cuneiform script):
Cuneiform script, one of the earliest systems of writing,
was invented by the Sumerians. It is distinguished by its
wedge-shaped marks on clay tablets, made by means of a blunt
reed for a stylus. The name "cuneiform" itself simply means
"wedge shaped".

(Autograph in Assyriology):
An autograph in Assyriology is the hand-copy of a cuneiform
clay-tablet. Producing an autograph is often the first step of
a tablet's archaeological interpretation and the autograph is
frequently the authoritative form that is published as source
material. Autographing the text is followed by transliteration,
transcription and translation.
```

**Answer:** Sumerians

---

## E LIMITATIONS

**Order-insensitivity of composition.** In COMPAS, parameter-space composition is strictly commutative and associative, since adapters are summed without regard to order. This differs from textual concatenation, where $(c_A, c_B)$ and $(c_B, c_A)$ may yield different interpretations. While our results show benefits of order-robustness (mitigating prompt-order sensitivity), the lack of positional information may reduce expressivity in tasks where sequence order is essential, such as instruction chaining or narrative reasoning. In additional experiments, we provide a proof of concept showing that positional information can be encoded within the generator, demonstrating feasibility but leaving systematic exploration to future work.

**Generator robustness.** Our generator is trained on a meta-distribution of contexts and shows cross-domain transfer, but generalization to entirely novel domains or reasoning styles is not guaranteed. In particular, low-resource or highly specialized tasks may expose brittle adaptation.

**Linearity vs. expressivity.** The framework enforces strict linear compositionality of adapters to ensure interpretability and additivity guarantees. However, some tasks inherently require nonlinear interactions between contexts (e.g., resolving contradictions or multi-hop reasoning). In such cases, COMPAS may face a trade-off between maintaining modularity and achieving task-optimal integration.

**Scalability and efficiency trade-offs.** Although COMPAS replaces long contexts with compact adapters, the generator itself introduces a large up-projection layer whose parameter count grows

with the size of the student adapters. This overhead is modest relative to the base LM but still significant. Scaling to even larger backbones or more complex adapter schemes may therefore incur efficiency and memory costs that partially offset the gains from shorter inference contexts.

## LLM USAGE DISCLOSURE

In preparing this manuscript, we made limited use of LLMs to polish the writing at the sentence level. This assistance was limited to stylistic improvements such as grammar, clarity, and conciseness.

