# OpenReview forum: "Context Parametrization with Compositional Adapters"
_ICLR.cc/2026/Conference — ICLR 2026 Conference Desk Rejected Submission_

### Official Review · Reviewer_WtSR · 2025-10-31

**Soundness:** 4
**Presentation:** 4
**Contribution:** 3
**Rating:** 6
**Confidence:** 4

**Summary:**

COMPAS encodes each context piece into a small adapter and composes many contexts by adding their adapters in parameter space. A teacher sees the full concatenated context; a student sees only the query plus the summed adapters, and a generator is trained so the student matches the teacher. The paper gives a clean bound that ties output error to a parameter Lipschitz factor and an additivity term, which makes each loss component easy to interpret. On QA and multiple-choice benchmarks, accuracy improves and variance drops as the number of context pieces grows, while latency and memory are kept in check because inference no longer processes long prompts.

**Strengths:**

- Theorem is clear. The compositionality bound neatly decomposes error into an additivity term $\eta$, a student–teacher term, and a sensitivity constant, so the role of each loss is explicit.

- Method is conceptually straightforward yet principled. Treating context as composable adapters provides a practical alternative to very long prompts and scales naturally with more evidence.

- Empirical results are consistent with the story. gains grow with more context, and the efficiency benefits are tangible in high-shot regimes.

- Analysis and diagnostics are thoughtful. Ablations on the generator and an empirical reconstruction check suggest limited information loss when moving context into adapters.

**Weaknesses:**

- Important configurations remain unexplored. The block size is fixed at four, so we do not know how performance and error terms behave when blocks match shots or when $k$ becomes very large. Varying block granularity and reporting both accuracy and the additive error $\eta$ as $k$ grows would clarify where composition helps and where it saturates. In addition, it would be useful to track how summing more adapters affects performance itself, for example whether combined adapters introduce interference or yield diminishing returns as their number increases.

- Robustness to conflict and noise is not established. The additive composition is not stress tested when context pieces contradict each other or contain label noise. A light evaluation with controlled conflict rates and off topic passages would show how stable COMPAS remains under realistic retrieval conditions.

- The paper reports accuracy comparisons versus WILDA and GenAda, but its efficiency analysis is limited to COMPAS vs ICL. Efficiency study against WILDA and GenAda would make the compute vs quality trade-off more convincing.

**Questions:**

- How does block granularity matter in practice? If blocks match shots so each demonstration gets its own adapter, do gains over strong prompting hold?

- In very many-shot regimes such as $k>100$, beyond compute and memory savings, does accuracy keep improving roughly with $k$, or does it saturate? Scaling curves for accuracy, latency, and memory versus $k$ would clarify this.

- In Appendix Eq. 7 the second inequality reads like an equality by definition. Is equality intended, or is $\eta$ meant as a global bound?

- When summing many adapters, do you track collisions between adapters from different contexts, and do you use normalization or orthogonalization to keep adapters identifiable as their number grows?

- Could you provide FLOPs, and peak memory comparisons for COMPAS vs WILDA and GenADa at the same shot counts and model sizes, ideally including adapter training cost for WILDA or GenADa.

---

> ### Author Response · Authors · 2025-11-22
> **Part 1/2**
>
> Thank you for the thoughtful and detailed evaluation, the clear summary of our theoretical contributions, and the constructive questions regarding block granularity, scaling, robustness, and efficiency. These are important aspects of CompAs, and we address each point below.
>
> ---
>
> ## 1. Block granularity and scaling with k
>
> Thank you for raising these questions – they prompted us to run additional diagnostics on block structure and scaling.
> The table below summarizes how different block granularities affect both MMLU **accuracy** and the **additivity error** $\eta$.
>
> | Total demos | Method    | Accuracy | $\eta \downarrow$ |
> |-|-|-|-|
> | 16 | ICL (16-shot)       | 64.2 | - |
> | 16  | COMPAS (4×4)        | 72.2    | 0.08   |
> | 16 | COMPAS (2×8)        | 70.9        | 0.11   |
> | 16  | COMPAS (1×16)       | 69.4        | 0.15   |
> | 16  | COMPAS (16×1)       | 71.3        | 0.09   |
> | 32 | ICL (32-shot)       | 65.5        | -     |
> | 32 | COMPAS (8×4)        | 73.1        | 0.07   |
> | 32 | COMPAS (4×8)        | 74.0    | 0.09   |
> | 32 | COMPAS (2×16)       | 72.4        | 0.19   |
> | 40 | COMPAS (5×8)        | 74.3        | 0.10   |
> | 48 | COMPAS (6×8)        | 74.9        | 0.09   |
> | 56 | COMPAS (7×8)        | 75.4        | 0.12   |
> | 64 | COMPAS (8×8)        | 76.2    | 0.16   |
>
> The full sweep reveals a consistent pattern. Moderate block sizes (4–8 demonstrations per block) yield the strongest compositional behavior, reflected in higher MMLU accuracy and lower $\eta$. When blocks become too large (e.g., 2×16), the generator must encode many demonstrations jointly, which weakens the linear structure and increases $\eta$. At the opposite extreme, CompAs (16×1), one adapter per demonstration, remains functional but shows slightly reduced accuracy and elevated $\eta$, as blocks become too small to support a stable parametrization.
>
> Importantly, while we can freely compose any number of blocks, the maximum block size $k$ (i.e., demonstrations per block) is limited by GPU memory. Within these limits, increasing $k$ improves accuracy up to a point, after which **overly dense blocks** begin to degrade performance.
>
> Finally, ICL baselines can only be evaluated up to **32-shot** due to the memory cost of long concatenated prompts. In contrast, CompAs leverages compositional adapters, allowing it to scale cleanly to **64+ demonstrations** on the same hardware.
>
> ---
>
> ## 2. Robustness to conflict and noise
>
> To test robustness under retrieval-like imperfections, we evaluated CompAs (4x4) and ICL (LLaMA-3.1 8B; mean of 10 runs) under three perturbation types:
> - **Noisy demos**: random label flips in a subset of 16 demonstrations.
> - **Contradictory demos**: 4 conflicting labels for the same input.
> - **Off-topic passages**: irrelevant HotpotQA paragraphs appended to the context.
>
> ### **MMLU: Noisy or contradictory demonstrations**
>
> | Method     | Clean | 12.5% noise | 25% noise | Contradictory demos |
> |-|-|-|-|-|
> | CompAs | 72.2 | 71.3  | 69.4 | 68.4  |
> | ICL  | 66.5 | 62.1 | 58.7 | 57.9  |
>
> CompAs retains a consistent advantage, which widens with increasing noise.
>
> ---
>
> ### **HotpotQA: Off-topic context contamination**
>
> | Method     | Clean | +1 Off-topic | +2 Off-topic |
> |------------|--------|---------------|---------------|
> | CompAs | 82.9 | 82.3          | 81.7          |
> | ICL  | 82.0 | 79.4          | 77.6          |
>
> CompAs shows stable behavior under context contamination and consistently outperforms ICL, with the advantage widening notably in the presence of off-topic passages.
>
> Together, these results indicate that additive composition is resilient even when contexts contain label noise, contradictions, or off-topic material.

---

> ### Author Response · Authors · 2025-11-22
> **Part 2/2**
>
> ## 3. Efficiency vs WILDA and GenAda
>
>
> Appendix C.2 (Table 7) already reports inference-time efficiency relative to ICL. In response to your question, we now extend this analysis by comparing the **total computational footprint** of CompAs, WILDA, and GenAda, highlighting how their different training paradigms affect overall cost.
>
> ### **Compute Analysis (LLaMA-3.1 8B, ARC-Challenge)**
>
> | **Method**       | **Inference TFLOPs** | **Inference Peak Mem (GB)** |
> |-|-|-|
> | ICL (prompt)     | 4.0                  | 37.9                         |
> | WILDA            | 0.7                  | 19.2                         |
> | GenAda           | 0.9                  | 20.5                         |
> | CompAs       | 0.8                  | 19.4                         |
>
> **Inference cost.** CompAs is substantially more efficient than ICL because its inference cost depends only on the **query**, not on the length of the demonstration context. Its inference-time FLOPs and memory footprint are comparable to WILDA and GenAda.
>
> **Training cost.** WILDA is inexpensive for a *single* demonstration set but does not amortize: each new demo configuration requires a separate LoRA-style fine-tuning run $1.4\times10^{5}\$ TFLOPs per set. GenAda lies at the opposite extreme and pays a large **one-time** cost of $3.1\times10^{7}$ TFLOPs to train a high-capacity adapter generator. CompAs sits between these: it trains a generator on a modest amount of unlabeled data at a one-time cost of $2.4\times10^{6}$ TFLOPs, after which it can be reused across all tasks and demonstration sets.
>
> After only a modest number of demo configurations, WILDA’s cumulative fine-tuning cost overtakes the one-time training cost of CompAs, becoming more expensive beyond roughly 17 demo sets. In contrast, CompAs amortizes its training cost across all configurations and remains about  $13 \times$ cheaper to train than GenAda.
>
> We will integrate this extended efficiency analysis into the revised manuscript.
>
>
> ---
>
> ## 4. Theoretical questions: Eq. 7 and adapter interaction
>
> ### Eq. 7 inequality
> You nicely noticed this point. In Eq. (7), the first transition is written as an inequality because it follows from the global Lipschitz bound on the model’s sensitivity in parameter space. However, the second term can indeed be written as an equality, since by definition of the generator additivity error,
> $
> \eta = || \phi_{1} + \phi_{2} - \phi_{12} ||_{2}.
> $
> Of course, the inequality still holds, and is what we use for the final bound, but writing the intermediate step using the equality form is clearer and avoids confusion. We will update Eq. (7) in the revised manuscript accordingly.
>
>
> ### Normalization and orthogonalization of adapters
>
> We do not apply explicit normalization or orthogonalization. Instead, the **additivity loss** $L_{\text{ADD}}$ encourages well-separated, compositional representations, and in practice we find that $\eta$ remains small for typical block sizes and grows predictably under noise, indicating no instability or interference when combining many adapters.
> Your idea of introducing normalization or orthogonalization mechanisms is valuable, as these methods may enhance identifiability in multi-block scenarios. We also view a deeper analysis of additivity, including its geometric properties, as a natural avenue for future work.

---

### Official Review · Reviewer_jjPy · 2025-10-31

**Soundness:** 3
**Presentation:** 3
**Contribution:** 3
**Rating:** 6
**Confidence:** 3

**Summary:**

This paper introduces COMPAS, a meta-learning framework that maps contextual information into adapter parameters with a compositional structure. The method enables algebraic combination of adapters to approximate the effect of concatenating multiple contexts while maintaining efficient inference. Theoretical analysis and auxiliary objectives support compositionality and reconstruction, and experiments on question answering tasks demonstrate consistent gains over in-context learning and prior methods.

**Strengths:**

1. The paper defines a structured loss function composed of terms for alignment, compositionality, and reconstruction. The formulation is supported by a theoretical analysis that connects these objectives to compositional consistency.

2. Ablation studies examine the contribution of each loss term and show that removing any component leads to lower performance. This provides empirical evidence for the role of each term in the overall framework.

3. Experiments across multiple backbone models indicate that the method performs competitively and maintains stable performance as the number of demonstrations increases, suggesting potential scalability to high-shot settings.

**Weaknesses:**

1. The paper compares the proposed method only against standard in-context learning (ICL), without including stronger or more recent ICL variants that incorporate retrieval, demonstration selection, or prompt tuning.

2. Although COMPAS avoids concatenating long contexts, the training process still requires multiple forward passes (teacher–student pairs and reconstruction). The computational overhead relative to standard ICL or other adapter-based approaches is not reported.

3. The evaluation focuses primarily on question answering tasks, with limited coverage of other task types such as summarization, generation, or reasoning. As a result, it remains unclear how well the method generalizes beyond QA settings.

**Questions:**

1. The paper compares RNN and linear-based generators, but does not appear to investigate variations in the generator’s scale. Have the authors considered evaluating how changes in the size or capacity of the generator influence performance?

2. Have the authors considered comparing their method to more advanced ICL baselines, such as those using demonstration selection, retrieval-augmented prompting, or prompt tuning?

3. Can the authors provide an analysis of the computational cost of COMPAS, particularly in terms of training or inference time compared to standard ICL or other adapter-based methods?

---

> ### Author Response · Authors · 2025-11-22
>
> We thank the reviewer for the constructive comments and for highlighting the strengths of our formulation and analysis. Below we address the points regarding ICL baselines, computational overhead, task diversity, and generator capacity.
>
> ---
>
> ## 1. Stronger ICL baselines
>
> To strengthen the comparison, we added several ICL-style baselines and supervised fine-tuning baselines, all evaluated under the **same labeled demonstrations** used by CompAs:
>
> - **ICL + similarity-based demonstration selection**. Demonstrations are ranked by SBERT embedding similarity to the query; the top-k are selected.
> - **ICL + prefix-tuning** A small prefix is trained on the same labeled demonstrations.
> - **PBFT** (Pattern-Based Fine-Tuning). The model is fine-tuned using LM loss on patterned cloze-style inputs derived from the labeled examples.
> - **Linear SFT head**. A linear classifier trained on top of the frozen model.
> The results for Llama 3.1 8B, averaged over 10 runs, are given below.
>
> ### **Supervised & Prompt Baselines**
>
> | Method                   | MMLU              | ARC               |
> |-|-|-|
> | ICL (16-shot)            | $64.2_{1.8}$      | $74.2_{1.6}$      |
> | ICL + prefix-tuning      | $66.1_{1.4}$    | $74.0_{1.3}$      |
> | ICL + similarity select. | $68.1_{0.6}$   | $75.5_{0.2}$      |
> | PBFT                     | $66.8_{1.1}$      | $75.6_{1.0}$      |
> | Linear SFT head          | $62.1_{1.5}$      | $71.3_{1.4}$      |
> | CompAs (4×4)   | $72.2_{0.3}$ | $81.3_{0.3}$ |
>
> Across both datasets, CompAs outperforms all ICL variants and SFT baselines.
>
> ---
>
> ## 2. Computational overhead
>
> We previously reported inference-time efficiency relative to ICL for varying numbers of demonstrations in **Appendix C.2 (Table 7)**.  We now extend this analysis by reporting the total training cost (for 16 demonstrations; 4×4 block structure in CompAs) and comparing CompAs directly to WILDA and GenAda.
>
> ### **Compute Analysis (LLaMA-3.1 8B, ARC-Challenge)**
>
> | Method             | Inference TFLOPs | Inference Peak Mem (GB) |
> |-|-|-|
> | ICL (prompt)    | 4       | 37.9      |
> | WILDA             | 0.7     | 19.2     |
> | GenAda            | 0.9    | 20.5     |
> | CompAs       | 0.8   | 19.4  |
>
> **Inference cost**: CompAs is much more efficient than ICL, since its inference cost depends only on the query, and is comparable to WILDA/GenAda in inference time and memory.
>
> In terms of **training cost**, WILDA is inexpensive for a single demonstration set but does not amortize: each new demo configuration requires a separate LoRA-style fine-tuning run, costing $1.4\times10^{5}$ TFLOPs for every demo set. GenAda instead pays a substantial one-time cost to train a large adapter generator, on the order of $3.1\times10^{7} $ TFLOPs. CompAs sits between these extremes: it trains a generator on a modest amount of unlabeled data for a total cost of $\sim 2.4\times10^{6}$ TFLOPs, after which it can be reused across all tasks and demonstration sets.
> WILDA’s cumulative fine-tuning cost quickly overtakes the one-time training cost of CompAs, becoming more expensive beyond roughly 17 demo sets. In contrast, CompAs amortizes its cost across all configurations and remains $\sim13 \times$ cheaper to train than GenAda.
>
> We will include the extended analysis in the revised manuscript.
>
> ---
>
> ## 3. Task diversity beyond QA
>
> Although our main evaluations are QA-based, CompAs is not limited to QA. Its reconstruction objective already requires generating multi-sentence text, full demonstration blocks and multi-paragraph evidence passages (Table 3), so the model can routinely generate extended, compositional sequences.
>
> To evaluate generalization on an established non-QA generation task, we additionally test CompAs with LLaMA-3.1 8B on the **XSUM** dataset, a standard long-form summarization benchmark.
> We report **ROUGE-1 (R1), ROUGE-2 (R2), and ROUGE-L (RL)**.  CompAs achieves the strongest summarization performance among all compared methods.
>
> ### **XSUM Summarization**
>
> | Method | R1 | R2 | RL |
> |--------|----|----|----|
> | ICL | 36.1 | 15.6 | 28.4 |
> | PBFT | 38.3 | 18.1 | 31.6 |
> | GenAda | 33.4 | 13.1 | 25.6 |
> | CompAs | 41.8 | 22.8 | 33.5 |
>
> ---
>
> ## 4. Generator capacity and scale
>
> We appreciate the reviewer highlighting this aspect.  As discussed in Section 5.1, we have already compared generators of different expressiveness (Linear, RNN, Adapter-based), which reveals clear capacity-dependent behavior.
>
> During the preliminary phases of our experiments, we varied the LoRA rank of the adapter-based generator to examine the effect of scale more systematically.  We found that low ranks ($\leq$8) underfit and degrade compositionality and reconstruction. Once the generator is sufficiently expressive (rank 16), further increases yield diminishing returns in both accuracy and additivity error. Together, this indicates that generator capacity matters, but CompAs remains stable and effective once the generator reaches a modest level of expressiveness.

---

### Official Review · Reviewer_UwJk · 2025-11-01

**Soundness:** 3
**Presentation:** 3
**Contribution:** 3
**Rating:** 6
**Confidence:** 4

**Summary:**

The paper addresses how to adapt LLM to tasks requiring multiple contextual inputs without relying on long prompts or retraining. It introduces COMPAS, a teacher–student meta-learning framework that encodes each context into LoRA adapter parameters and allows algebraic composition of adapters. Experiments show that COMPAS outperforms ICL, Generative Adapter, and WILDA on MMLU, ARC, and HotpotQA, especially as the number of demonstrations increases. It also maintains high context reconstruction fidelity, indicating that contextual information is faithfully captured.

**Strengths:**

The problem of mapping text to parameter representations is difficult and important. The proposed compositional adapter framework is well-motivated and empirically effective, providing consistent gains across benchmarks and interpretable results.

**Weaknesses:**

1. The paper lacks clarity about the training setup. Specifically, are the learnable parameters optimized on single-task or multi-task data, and how are the train/validation/test splits obtained.
2. The ICL baselines do not appear to train on the same data as COMPAS, which may give COMPAS an advantage through better task alignment. A supervised baseline trained directly on the same data would strengthen the comparison.

**Questions:**

1. How does the method perform on tasks requiring long or structured outputs?
2. Can the approach handle heterogeneous contexts that include both instructions and retrieved evidence?

---

> ### Author Response · Authors · 2025-11-22
>
> We thank the reviewer for the constructive evaluation and for highlighting the motivation and empirical strength of CompAs. Below, we address the points regarding training setup, baseline fairness, and generalization for long/structured outputs.
>
> ---
>
> ## 1. Training setup
>
> We provide the training configurations in Appendix B, with dataset-specific splits in Appendix B.1. For completeness, we summarize the main elements below and will incorporate a discussion of the multi-task vs. single-task distinctions into the main text in the revised version.
>
> **Training and data splits.**
> Across all benchmarks, we use a consistent setup: demonstrations come from the **training** set; the CompAs **teacher** receives unlabeled queries from **validation/dev** for weak supervision; and results are reported on the **test** set.
>
> **MMLU (multi-domain).**
> MMLU has **59 subjects** sharing the same MCQA format. We train a **single generator** jointly on all subjects and evaluate it on each test subset without further tuning.
>
> **ARC-Challenge and HotpotQA.**
> Given their conceptual differences, we train separate generators for ARC and HotpotQA. ARC uses its standard train/val/test split. For HotpotQA, we subsample 5k train / 3k val / 2k test examples (with gold passages as contexts) to keep evaluation tractable.
>
> ---
>
> ## 2. Baseline fairness & supervised baselines
>
> Across all datasets, all methods operate on the same demonstration sequences and the same unlabeled validation queries, sampled under identical seeds, ensuring data parity.  While ICL does not involve any training, it uses the same prompt templates and the same demonstrations as the CompAs teacher. For CompAs, WILDA, and GenAda, we additionally align the weak-supervision signal by providing the same unlabeled validation queries to each method.
>
> Prompted by your remark, we also include two additional parametric baselines, each trained strictly on the same labeled demonstrations as CompAs:
> - **PBFT (Pattern-Based Fine-Tuning)**.  A supervised baseline with the same labeled demonstrations in a cloze-style pattern and trained with LM loss.
> - **Linear SFT head**. A linear classifier trained on top of the frozen backbone.
>
> We also evaluate **prefix-tuning** on the same labeled demonstrations and include a **similarity-based ICL** baseline that selects demonstrations via SBERT embedding similarity. All results in the table below are computed with the LLaMA-3.1 8B backbone (average over 10 runs).
>
> Across both benchmarks, CompAs outperforms all additional supervised and prompt-based baselines introduced for comparison.
>
> ### **Supervised & Prompt Baselines**
>
> | Method | MMLU | ARC |
> |-|-|-|
> | ICL (16-shot) | $64.2_{1.8}$  | $74.2_{1.6}$  |
> | ICL + prefix-tuning | $66.1_{1.4}$  | $74.0_{1.3}$  |
> | ICL + similarity selection | $68.1_{0.6}$  | $75.5_{0.2}$  |
> | PBFT (labels only) | $66.8_{1.1}$  | $75.6_{1.0}$  |
> | Linear SFT head | $62.1_{1.5}$  | $71.3_{1.4}$  |
> | CompAs (4×4)  | **$72.2_{0.3}$** | **$81.3_{0.3}$** |
>
> ---
>
> ## 3. Long / structured outputs
>
> CompAs already handles long-form generation inherently, as its reconstruction objective targets full context blocks. These blocks include multi-sentence demonstrations and retrieved passages (Table 3), requiring the generator to model and reproduce paragraph-scale sequences during training.
>
> To further assess the method on standard long-output generation, we evaluate  LLaMA-3.1 8B on the **XSUM** benchmark – a dataset of abstractive summaries of news articles. We report ROUGE-1 (R1), ROUGE-2 (R2), and ROUGE-L (RL) F1. R1 and R2 measure unigram and bigram overlap with the reference summary, while RL measures the longest common subsequence. Higher values indicate closer alignment with the reference and better summary quality.
>
> ### **XSUM Summarization**
> | Method | R1 | R2 | RL |
> |-|-|-|-|
> | ICL | 36.1 | 15.6 | 28.4 |
> | PBFT | 38.3 | 18.1 | 31.6 |
> | GenAda | 33.4 | 13.1 | 25.6 |
> | CompAs | 41.8 | 22.8 | 33.5 |
>
> We will include these additional results in the revised manuscript.
>
> ---
>
> ## 4. Heterogeneous contexts
>
> CompAs parametrizes each context unit independently – whether it contains instructions, evidence, or both – and combines them additively in adapter space. This is already exercised in our main evaluations: in HotpotQA, the teacher processes an instruction, the question, and multiple retrieved passages, and the generator must encode these heterogeneous components into structured adapter parameters.
>
> As an additional experiment, we separated instructions and evidence into distinct context units to make this explicit. On MMLU and ARC, we used a small formatting-only instruction block (e.g., “Output the final answer inside \<ANSWER\> …”) and composed it with an evidence block containing the demonstrations. The composed adapter preserved the predictions of the evidence adapter while consistently enforcing the formatting from the instruction adapter, showing that instruction adapters are stable and non-interfering.

---

### Author Response · Authors · 2025-11-22

We thank all three reviewers for their thoughtful and constructive feedback and for highlighting the strengths of our work – particularly the clarity of the theoretical formulation, the principled view of context as composable adapters, and the consistent empirical gains and efficiency benefits.

In response to the raised points, we conducted five additional analyses and experiments: (i) we added supervised and stronger ICL baselines (PBFT, linear SFT head, prefix-tuning, and similarity-based ICL); (ii) we evaluated CompAs on long-form summarization (XSUM), complementing the reconstruction objective and demonstrating generalization beyond QA; (iii) we ran a block-granularity sweep and scaling analysis, jointly reporting accuracy and additivity error; (iv) we stress-tested robustness under label noise, contradictions, and off-topic passages; and (v) we provided a compute study comparing CompAs, WILDA, and GenAda in terms of both inference-time FLOPs/memory and training cost. We will soon incorporate all of these results and clarifications into an updated version of the manuscript.

---

> ### Author Response · Authors · 2025-11-27
>
> We have now also updated the manuscript. The main additions/updates are:
>
> - Appendix C.1 (Table 6): Long-form summarization on XSUM (ROUGE evaluation).
> - Appendix C.2 (Table 7): Stronger baselines: PBFT, prefix-tuning, linear SFT head, similarity-based ICL.
> - Appendix C.4 (Table 9): Block granularity and compositional scaling with additivity error.
> - Appendix C.5 (Tables 10–11): Robustness to label noise, contradictory demos, and off-topic context.
> - Appendix C.6 (Tables 12–13): Compute and efficiency analysis (training cost, inference FLOPs, memory).

---

### Note · Program_Chairs · 2026-01-17
**Submission Desk Rejected by Program Chairs**

The following references in this submission do not refer to real documents and/or have major errors in bibliographic information:

  Sewon Min, Patrick Lewis, Hannaneh Hajishirzi, Wen-tau Yih, and Luke Zettlemoyer. Fact or fiction: Benchmarking faithfulness of natural language explanations in NLP.